# Nitrogen and CO$_2$ enrichment interact to decrease biodiversity impact on complementarity and selection effects

Mengjiao Huang [1,2] ✉, Peter B. Reich [3,4,5], Shaopeng Wang [6], Yanhao Feng [7], Pubin Hong [6], Kathryn E. Barry [8], Miao He [8,9], Shengman Lyu[10], Shurong Zhou [11], Neha Mohanbabu[3], Forest Isbell [9] & Yann Hautier [8]

Global environmental change is causing a decline in biodiversity with profound implications for ecosystem functioning and stability. It remains unclear how global change factors interact to influence the effects of biodiversity on ecosystem functioning and stability. Here, using data from a 24-year experiment, we investigate the impacts of nitrogen (N) addition, enriched CO$_2$ (eCO$_2$), and their interactions on the biodiversity-ecosystem functioning relationship (complementarity effects and selection effects), the biodiversity-ecosystem stability relationship (species asynchrony and species stability), and their connections. We show that biodiversity remains positively related to both ecosystem productivity (functioning) and its stability under N addition and eCO$_2$. However, the combination of N addition and eCO$_2$ diminishes the effects of biodiversity on complementarity and selection effects. In contrast, N addition and eCO$_2$ do not alter the relationship between biodiversity and either species asynchrony or species stability. Under ambient conditions, both complementarity and selection effects are negatively related to species asynchrony, but neither are related to species stability; these links persist under N addition and eCO$_2$. Our study offers insights into the underlying processes that sustain functioning and stability of biodiverse ecosystems in the face of global change.

The Earth is currently experiencing an unprecedented decline in biodiversity due to global environmental changes[1,2]. This loss of biodiversity can have significant impacts on the functioning of ecosystems and their ability to reliably provide essential functions to humanity[3–7].

Therefore, to develop effective management strategies that optimize ecosystem functioning and stability[8–10], it is crucial to understand the processes driving biodiversity loss in response to global change, as well as the resulting consequences on ecosystem functioning and stability.

[1]National Observation and Research Station for Shanghai Yangtze Estuarine Wetland Ecosystems, and Ministry of Education Key Laboratory for Biodiversity Science and Ecological Engineering, Institute of Biodiversity Science, School of Life Sciences, Fudan University, Shanghai, China. [2]Basque Centre for Climate Change (BC3), Leioa, Spain. [3]Department of Forest Resources, University of Minnesota, Saint Paul, MN, USA. [4]Hawkesbury Institute for the Environment, Western Sydney University, Penrith, NSW, Australia. [5]Institute for Global Change Biology, School for Environment and Sustainability, University of Michigan, Ann Arbor, MI, USA. [6]Institute of Ecology, College of Urban and Environmental Science, and Key Laboratory for Earth Surface Processes of the Ministry of Education, Peking University, Beijing, China. [7]State Key Laboratory of Herbage Improvement and Grassland Agro-ecosystems, College of Pastoral Agriculture Science and Technology, Lanzhou University, Lanzhou, China. [8]Ecology and Biodiversity Group, Department of Biology, Utrecht University, Utrecht, The Netherlands. [9]Department of Ecology, Evolution and Behavior, University of Minnesota, Saint Paul, MN, USA. [10]Department of Ecology and Evolution, University of Lausanne, CH-1015, Lausanne, Switzerland. [11]School of Ecology, Hainan University, Haikou, PR China. ✉e-mail: mjhuang17@fudan.edu.cn

Over the past decades, ecologists have conducted numerous experiments to understand the impacts of biodiversity on ecosystem productivity[6,11–13] and the stability of that productivity over time (mean/SD)[5,14–17]. Biodiversity can promote community productivity through two general classes of processes: complementarity effects due to resource partitioning, abiotic facilitation, and biotic feedbacks[18,19], and selection effects occur when highly productive species contribute disproportionally to community productivity[13]. Biodiversity can also promote community stability through two types of processes: species asynchrony, due to differential species responses to fluctuating environments or compensatory dynamics among species through time[17,20], and species stability, the average stability of all species in the community weighted by their abundances[21].

These bipartite frameworks help clarify the links between biodiversity-functioning and biodiversity-stability relationships, enhancing our knowledge of biodiversity theory and its application to ecosystem management[8,22–25]. Theoretical and empirical studies also suggest that complementarity and selection effects can be related to species asynchrony and species stability, and ultimately to community stability[22–27]. For example, complementarity effects can be either positively related to species asynchrony due to resource partitioning in time, or negatively related to species asynchrony when they vary with interspecific competition in opposite directions[26] (Table 1). While these studies have demonstrated the interconnected nature of the processes underlying biodiversity-functioning and biodiversity-stability relationships, to what extent global change affects these processes and their linkages remains unknown. Global change could

disrupt the relationships between processes driving productivity and stability, leading to ecosystems that are productive but unstable or stable but with low productivity. This misalignment complicates ecosystem management, emphasizing the need to understand and address these impacts.

Nitrogen (N) addition and $CO_2$ enrichment (e$CO_2$) are two key global change factors known to affect the effects of biodiversity on ecosystem functioning and stability[28] and may potentially cause the mismatch between the processes underlying biodiversity-functioning and biodiversity-stability relationships individually or interactively[29,30]. First, N addition can weaken the positive relationship between biodiversity and a relative measure of complementarity effects, diminish the positive contribution of biodiversity to species asynchrony, or strengthen the relationship between biodiversity and selection effects (Table 1). Second, e$CO_2$ can increase plant biomass[31,32] and temporal stability[33,34], or have no impact on productivity[30] or stability[9,35], as well as no direct impact on complementarity or selection effects[30,36]. Third, N addition and e$CO_2$ may have synergistic, antagonistic, or additive effects on productivity[28,37,38] and temporal stability. A combination of e$CO_2$ and N addition increased productivity more than e$CO_2$ alone[28,37,38]. However, whether e$CO_2$ and N addition interact to affect the processes underlying biodiversity-stability relationships, as well as biodiversity-functioning and biodiversity-stability relationships, remains unknown.

Here, using data from a 24-year experiment crossing treatments of species richness, N addition, and e$CO_2$, we aim to explore the impacts of N addition, e$CO_2$, and their interactions on the biodiversity-ecosystem functioning relationship, the biodiversity-stability

**Table 1 | Relationships between processes underlying the biodiversity-functioning relationship (complementarity and selection effects), the biodiversity-stability relationship (species asynchrony and species stability), and their relationships in the conditions of ambient and nitrogen (N) addition shown by previous studies**

| Relationship | Hypotheses and mechanisms |
| --- | --- |
| ***Ambient*** | |
| Species richness → Complementarity effects | Higher species richness leads to a higher complementarity effect due to resource partitioning, facilitation, and biotic feedbacks[13,18,19]. |
| Species richness → Selection effects | Higher species richness often leads to a negative selection effect because species with higher-than-average biomass in monoculture that perform relatively poorly in mixture[69], or a positive selection effect because species with a lower-than-average biomass in monoculture that are more abundant in mixture than expected[47]. |
| Species richness → Species asynchrony | Higher species richness leads to higher species asynchrony due to a greater likelihood for asynchronous fluctuations among species[20,29]. |
| Species richness → Species stability | Higher species richness can either lead to higher species stability (usually in observational experiments) due to mean population sizes of the natural communities increased with species richness and populations in species-poor communities may exhibit greater fluctuations in abundance than populations in species-rich communities, or lower species stability (usually in diversity-manipulated experiments) due to usually equal population sizes among species in communities[50,70]. |
| Complementarity effects → Species asynchrony | Within species richness levels, the complementarity effect can be either positively related to species asynchrony when both of them increase with resource partitioning[24,26], or negatively related to species asynchrony due to decreased complementarity effect but increased species asynchrony with competition[26]. |
| Complementarity effects → Species stability | Within species richness levels, the complementarity effect can be positively related to species stability due to increased species abundance and buffered demographic stochasticity with over-yielding effects[27,71]. |
| Selection effects → Species asynchrony | Within species richness levels, selection effect can be negatively related to species asynchrony due to increased dominance by particular species[21,71]. Selection effects can be positively related to species asynchrony[24], depending in part on whether the most productive species are also the most stable species[23]. |
| Selection effects → Species stability | Within species richness levels, selection effect can be either positively or negatively related to species stability due to higher monoculture productivity of the dominant species in the mixture are stable or not[23,71]. |
| ***N addition*** | |
| Species richness → Complementarity effects | N addition can weaken the positive relationship between diversity and the complementarity effect because of reduced positive species interactions, reduced niche dimensionality of soil nutrients[72]. |
| Species richness → Selection effects | N addition may weaken the negative relationship between diversity and selection effect or strengthen the positive relationship between diversity and selection effect because N addition increases competition for light, leading to increased dominance and reduced evenness[73]. |
| Species richness → Species asynchrony | N addition can weaken the positive relationship between diversity and species asynchrony because of reduced differential responses of species to environmental fluctuations, reduced belowground competition among species for nutrients, or reduced demographic stochasticity[42,43]. |
| Species richness → Species stability | N addition can also strengthen the negative relationship between diversity and species stability because of reduced competition among species for nutrients[43]. |

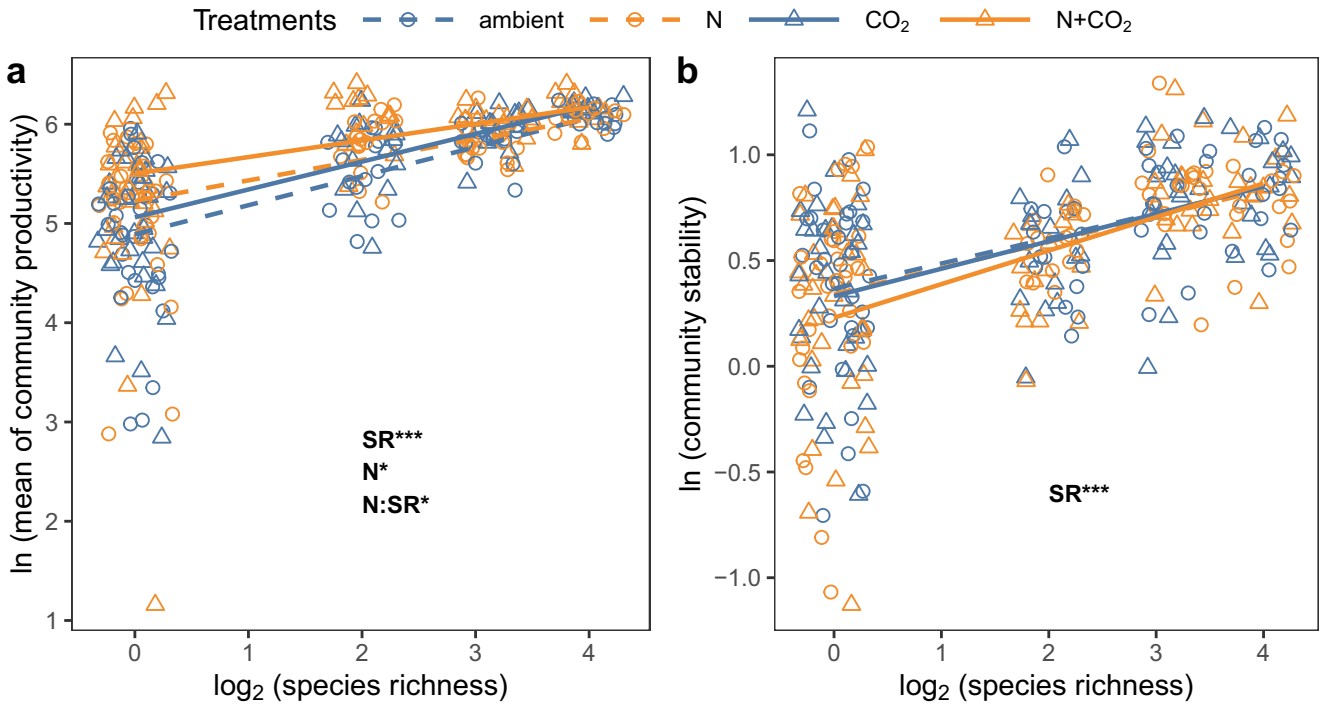

**Fig. 1 | Effects of CO₂ enrichment and nitrogen (N) addition on species richness-productivity and species richness-stability relationships.** Linear mixed-effects models followed by type III ANOVA for the effects of CO₂ enrichment and N addition on the relationships **a** between planted species richness (SR) and mean of community productivity; **b** between planted species richness (SR) and community stability, with rings as the random intercept. ***$P < 0.001$, **$P < 0.01$, *$P < 0.05$, .$P < 0.1$. The significant predictors are indicated by text in the figures. All tests were two-sided.

**Table 2 | Statistics results of linear mixed-effects models using the Anova function (with type III SS) for the effect of CO₂ enrichment (eCO₂), nitrogen (N) addition, and species richness on community stability, mean of community productivity, species asynchrony, species stability, complementarity effect, and selection effect across 24 years. Community stability, mean of community productivity, species asynchrony, species stability, and species richness were log-transformed. Significance of the fixed effects was assessed using two-sided log-likelihood ratios ($\chi^2$ values)**

| Predictors | Mean of community productivity | | | Community stability | | | Complementarity effect | | | Selection effect | | | Species asynchrony | | | Species stability | | |
|---|---|---|---|---|---|---|---|---|---|---|---|---|---|---|---|---|---|---|
| | $\chi^2$ | df | P value | $\chi^2$ | df | P value | $\chi^2$ | df | P value | $\chi^2$ | df | P value | $\chi^2$ | df | P value | $\chi^2$ | df | P value |
| SR | 99.64 | 1 | **<0.001** | 26.20 | 1 | **<0.001** | 55.26 | 1 | **<0.001** | 1.94 | 1 | 0.164 | 43.48 | 1 | **<0.001** | 8.27 | 1 | **0.004** |
| CO₂ | 1.33 | 1 | 0.249 | 0.09 | 1 | 0.764 | 1.06 | 1 | 0.303 | 0.02 | 1 | 0.895 | 1.11 | 1 | 0.292 | 1.60 | 1 | 0.206 |
| N | 5.29 | 1 | **0.021** | 0.07 | 1 | 0.794 | 3.82 | 1 | **0.050** | 0.43 | 1 | 0.513 | 1.07 | 1 | 0.301 | <0.01 | 1 | 0.978 |
| SR:CO₂ | 0.09 | 1 | 0.758 | 0.11 | 1 | 0.743 | 1.19 | 1 | 0.275 | 0.48 | 1 | 0.488 | 0.85 | 1 | 0.357 | 1.25 | 1 | 0.264 |
| SR:N | 4.31 | 1 | **0.038** | 0.02 | 1 | 0.896 | 8.10 | 1 | **0.004** | 1.78 | 1 | 0.182 | 1.24 | 1 | 0.265 | 0.03 | 1 | 0.856 |
| CO₂:N | 0.25 | 1 | 0.617 | 0.29 | 1 | 0.589 | 1.98 | 1 | 0.159 | 2.46 | 1 | 0.117 | 0.90 | 1 | 0.343 | 0.58 | 1 | 0.445 |
| SR:CO₂:N | 0.21 | 1 | 0.648 | 0.28 | 1 | 0.597 | 3.26 | 1 | **0.071** | 6.25 | 1 | **0.012** | 1.07 | 1 | 0.301 | 0.29 | 1 | 0.593 |

Bold values indicate statistically significant effects at $P < 0.1$.

relationship, as well as the links between their underlying processes by asking how N addition and eCO₂ impact:

(i) The effects of biodiversity on ecosystem productivity via complementarity and selection effects.
(ii) The effects of biodiversity on ecosystem stability via species asynchrony and species stability.
(iii) The links of complementarity and selection effects with species asynchrony and species stability.

## Results

### Processes and links underlying biodiversity-functioning and biodiversity-stability relationships under ambient conditions

Our bivariate analyses revealed that species richness increased community productivity ($\chi^2 = 99.64$, df = 1, $P < 0.001$, Fig. 1a; Table 2) by increasing complementarity effects ($\chi^2 = 55.26$, df = 1, $P < 0.001$, Fig. 2a; Supplementary Table S1), but did not alter selection effects ($\chi^2 = 1.94$, df = 1, $P = 0.164$, Fig. 2b). Species richness increased community stability ($\chi^2 = 26.20$, df = 1, $P < 0.001$, Fig. 1b; Table 2) by increasing species asynchrony ($\chi^2 = 43.48$, df = 1, $P < 0.001$, Fig. 2c) despite decreased species stability ($\chi^2 = 8.27$, df = 1, $P = 0.004$, Fig. 2d). Additionally, the bivariate analyses revealed a negative relationship between selection effects and species asynchrony ($\chi^2 = 4.61$, df = 1, $P = 0.041$, Fig. 3b; Supplementary Table 2), but no relationship between species stability and complementarity effects nor between species stability and selection effects ($\chi^2 = 0.62$, df = 1, $P = 0.430$, Fig. 3c; $\chi^2 = 0.58$, df = 1, $P = 0.447$, Fig. 3d). These results were further supported by structural equation models (SEM; Fig. 4a). However, although the bivariate analyses showed no relationship between complementarity effects and species asynchrony ($\chi^2 = 1.52$, df = 1, $P = 0.218$, Fig. 3a), the SEM analyses showed a negative relationship between them (Fig. 4a).

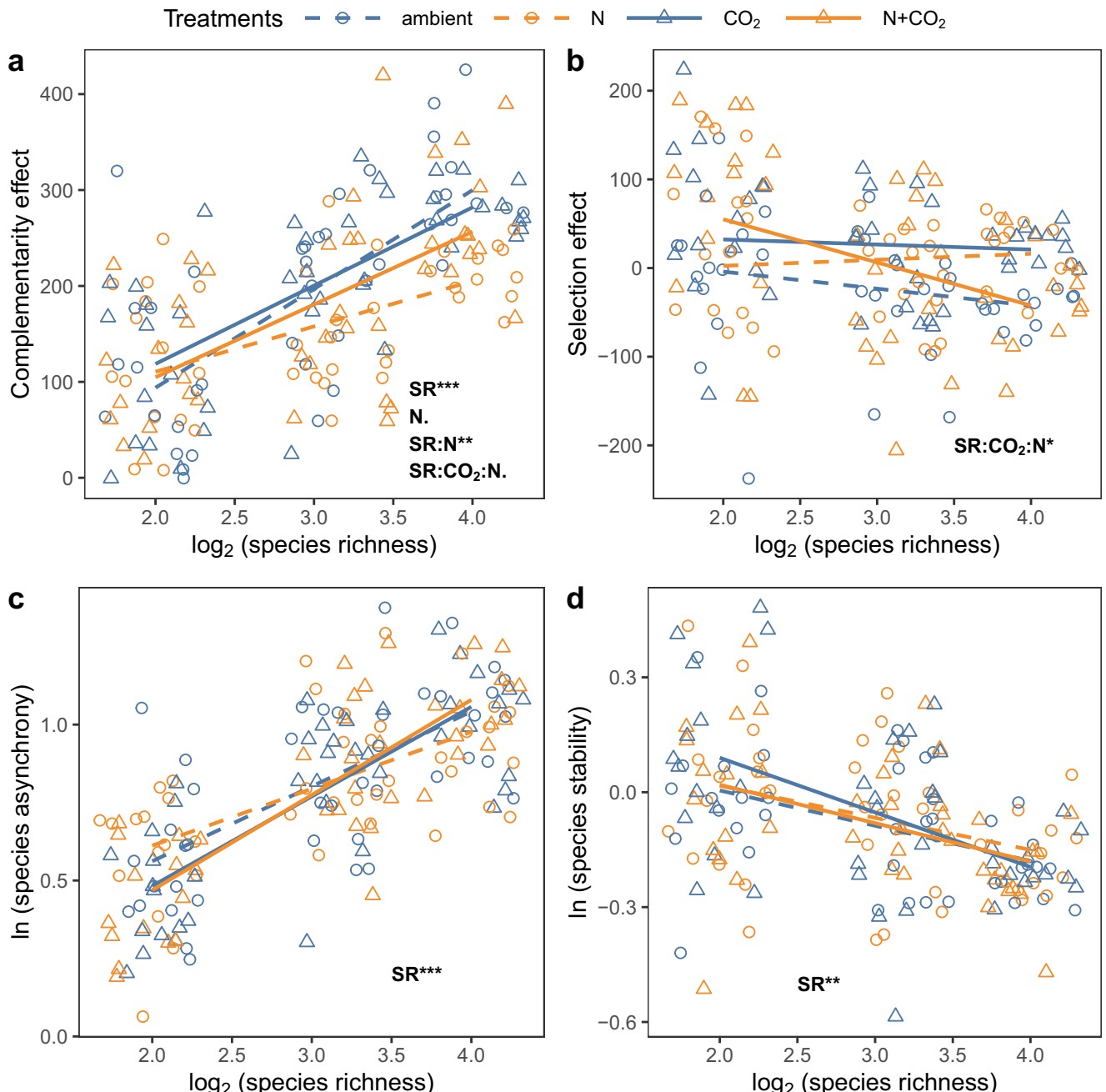

**Fig. 2 | Effects of CO$_2$ enrichment and nitrogen (N) addition on the processes underlying species richness-productivity and species richness-stability relationships.** Linear mixed-effects models followed by type III ANOVA for the effects of CO$_2$ enrichment and N addition on the relationships between planted species richness (SR) and **a** complementarity effect (CE), **b** selection effect (SE), **c** species asynchrony, and **d** species stability, with rings as the random intercept. ***$P < 0.001$, **$P < 0.01$, *$P < 0.05$, .$P < 0.1$. The significant predictors are indicated by text in the figures. All tests were two-sided.

## Impacts of N addition, eCO$_2$, and their interaction on biodiversity-ecosystem functioning and biodiversity-stability relationships

N addition increased community biomass more at lower than at higher species richness levels ($\chi^2 = 4.31$, df = 1, $P = 0.038$, Fig. 1a; Table 2). This is because N addition decreased complementarity effects more at higher species richness levels in both bivariate and SEM analyses ($\chi^2 = 8.10$, df = 1, $P = 0.004$, Figs. 2a; and 4c), although complementarity effects had a stronger positive relationship with productivity under N addition ($\chi^2 = 4.97$, df = 1, $P = 0.025$, Supplementary Fig. S1a; Fig. 4c; Supplementary Table S2). eCO$_2$ had no effect on the relationship between species richness and complementarity effects in bivariate analyses ($\chi^2 = 1.19$, df = 1, $P = 0.275$,

Fig. 1a), but weakened their positive relationship and also the positive relationship between complementarity effect and productivity in SEM (Fig. 4b). Both bivariate and SEM analyses showed that N addition and eCO$_2$ interacted to weaken the positive relationship between species richness and complementarity effects ($\chi^2 = 3.26$, df = 1, $P = 0.071$, Figs. 2a; and 4d), and the negative relationship between species richness and selection effects ($\chi^2 = 6.25$, df = 1, $P = 0.012$, Figs. 2b; and 4d). The relationships of species richness with community stability, species asynchrony, or species stability were not affected by N addition, eCO$_2$, or their interaction (Figs. 1; 2a, b; and 4c, d), while the positive relationships between species asynchrony and community stability were slightly different under different treatments in SEM (Fig. 4).

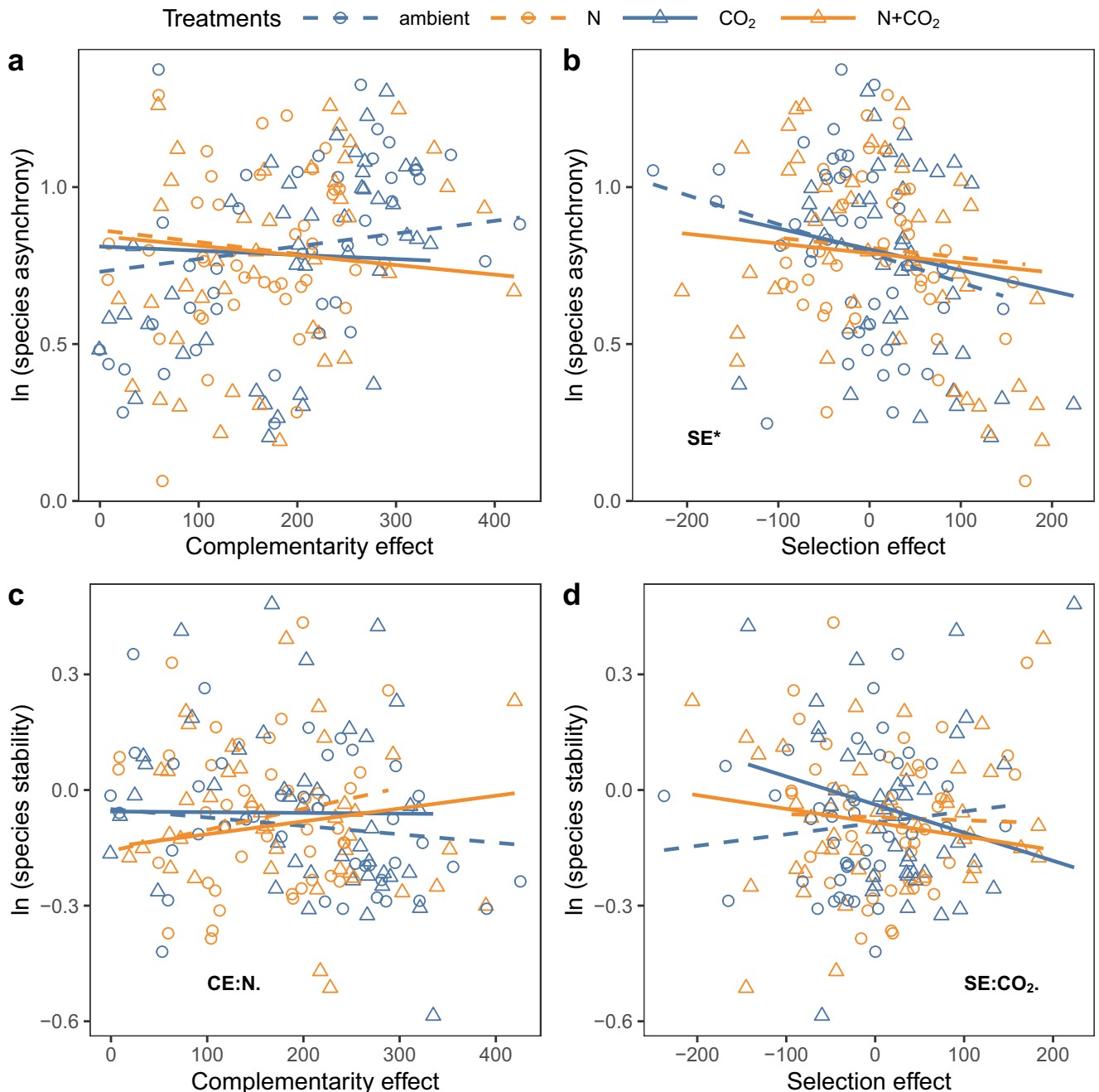

**Fig. 3 | Effects of $CO_2$ enrichment and nitrogen (N) addition on the links between species richness-productivity and species richness-stability relationships.** Linear mixed-effects models followed by type III ANOVA for the effects of $CO_2$ enrichment and N addition on the relationships **a** between species asynchrony and complementarity effect (CE); **b** between species asynchrony and selection effect (SE), **c** between species stability (spp_stab) and complementarity effect (CE); and **d** between species stability (spp_stab) and selection effect (SE), with species richness nested in rings as the random intercept. ***$P < 0.001$, **$P < 0.01$, *$P < 0.05$, .$P < 0.1$. The significant predictors are indicated by text in the figures. All tests were two-sided.

Moreover, bivariate relationships showed that N addition strengthened the relationship between complementarity effects and species stability, making it marginally positive ($\chi^2 = 2.89$, df = 1, $P = 0.089$, Fig. 3c; Supplementary Table S2). In contrast, eCO$_2$ weakened the relationship between selection effects and species stability, making it marginally negative ($\chi^2 = 3.17$, df = 1, $P = 0.075$, Fig. 3d). However, SEM showed that N addition, eCO$_2$, or their interaction had no detectable effect on the non-significant relationship between species stability and complementarity or selection effects, or the negative relationships of species asynchrony with complementarity or selection effects (Fig. 4).

## Discussion

Using data from the long-term BioCON experiment with treatments of biodiversity crossed with global change factors, we quantified the impact of two global change factors—nitrogen (N) addition and elevated $CO_2$ (eCO$_2$)—and their interaction on biodiversity-functioning relationships, biodiversity-ecosystem stability relationships, and links between these relationships. We found that N addition and eCO$_2$ interacted to decrease the effect of biodiversity on complementarity and selection effects. In contrast, N addition and eCO$_2$ had no effect on the relationship of biodiversity with species asynchrony and species stability, as well as on the links between biodiversity-functioning and

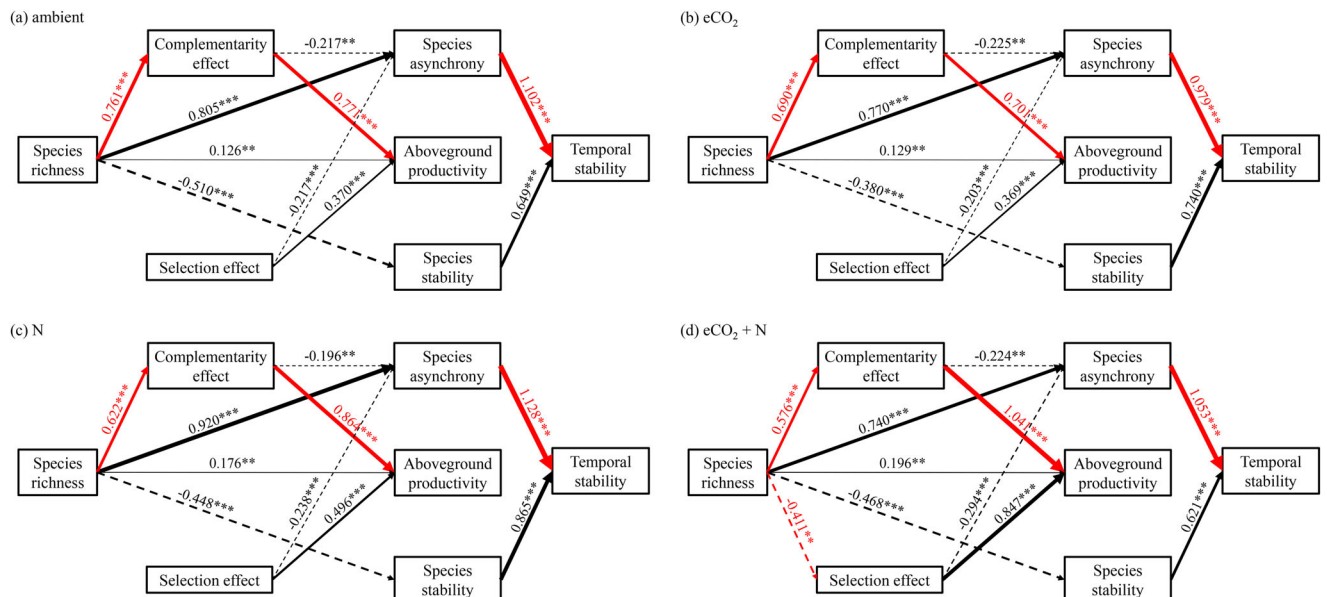

**Fig. 4 | Linking biodiversity effects and stability under different treatments.** Results of the multi-group structural equation model showing the relationships between mechanisms underlying biodiversity effects on ecosystem functioning and stability under different treatments: **a** ambient, **b** $CO_2$ enrichment (e$CO_2$), **c** nitrogen (N) addition, and **d** interaction between e$CO_2$ and N addition, from 1998 to 2021 (24 years). Black arrows represent significantly constrained paths ($P < 0.1$) while red arrows represent significant unconstrained paths between global change treatments. Solid and dashed arrows represent positive and negative paths, respectively. Non-significant paths are not shown ($P \geq 0.1$). Numbers on arrows are standardized path coefficients (scaled by their mean and standard deviation), and asterisks indicate statistical significance (***$P < 0.001$; **$P < 0.01$; *$P < 0.05$, .$P < 0.1$). The thickness of the arrows represents the strength of the path. Overall model: Fisher's $C = 16.112$, $P = 0.445$, AIC = 80.112, d.f. = 16.

biodiversity-stability relationships. While previous studies have separately quantified the effect of global change drivers on productivity[30,39–41], stability[10,42–45], or the links between the processes underlying biodiversity-functioning and biodiversity-stability relationships under ambient conditions[8,22–25], our study provides important insights into how N addition and e$CO_2$ interactively affect these links.

### Biodiversity-functioning and biodiversity-stability relationships under ambient conditions

Species richness increased community productivity by increasing complementarity effects while selection effects remained the same across the species richness gradient (Fig. 2a, b; Supplementary Fig. S1a, b). This result is consistent with many previous studies based on this experiment[46] and other experiments[25,47–49], indicating that complementarity effects have larger contributions to the stimulating effects of biodiversity on ecosystem functioning. In this experiment, complementarity effects increase productivity as a result of increasing functional trait diversity related to N cycling and availability[46]. However, there is considerable evidence that many different mechanisms may drive these complementarity effects in this system and others, such as resource partitioning, abiotic facilitation, and biotic feedbacks[19]. In line with previous studies[5,15,22,25,42,50], our study also shows that species richness increased community stability largely through increasing species asynchrony rather than species stability (Supplementary Fig. S1c, d; Fig. 2c, d; Fig. 4a).

Both SEM results (Fig. 4) and partial regression plots (after controlling the interactive effect between species richness, N addition, and e$CO_2$; Supplementary Fig. S2) showed a negative relationship between complementarity effects and species asynchrony. In contrast to this, Yan et al.[24] found that complementarity effects and species asynchrony were positively related. One possible reason is that the relationship between complementarity effects and species asynchrony may change with species richness. Indeed, performing our analysis within each species richness level revealed a positive relationship between

complementarity effects and species asynchrony at the species richness level of 4, but higher species richness levels tended towards negative relationships (Supplementary Fig. S3). Species richness may affect the relationships due to its impacts on competition between species[26]. The negative correlations among species in 16-species mixtures are weaker than in mixtures with fewer species (Supplementary Fig. S4). The reason is likely that interspecific competition in high-diversity communities drives communities towards having species with more distinct niches, which minimizes niche overlap and reduces competition among species[51], thereby decreasing species asynchrony (Supplementary Fig. S3). This change in species asynchrony may alter its relationship with complementarity effects.

### Biodiversity-functioning and biodiversity-stability relationships under N addition and e$CO_2$

Our SEM showed that N addition and e$CO_2$ affected the processes underlying the biodiversity-functioning relationship, but had no effects on the processes underlying the biodiversity-stability relationship or the links between processes underlying the two relationships. A potential explanation is that species richness may be a more important driver of biodiversity-stability relationships and the links between the two relationships, outperforming the effects of N addition or e$CO_2$ in bivariate models or SEM. Moreover, species richness may have greater impacts on competition-driven niche differentiation (corresponding to complementary effects and species asynchrony relationship) or species dominance (corresponding to selection effects and species asynchrony relationship) compared to N addition or e$CO_2$. This result suggests that N addition and e$CO_2$ do not alter the interconnection between the drivers of community productivity and stability. Additionally, it indicates that managing species richness is crucial for maintaining ecosystem functioning and stability, especially in the face of global change that may otherwise create a mismatch between these critical processes of ecosystem functioning and stability.

N addition weakened the positive effects of species richness on complementarity effects, and thereby the effects of species richness

on productivity. This result is consistent with results from a synthesis study combining 15 grassland biodiversity experiments[41]. The potential explanation may be that N addition reduced the dominance of legumes at higher species richness levels and associated N fixation, leading to reduced facilitative effects of legumes on other species in communities (Supplementary Fig. S6). We also found that $eCO_2$ slightly relieved the weakened positive relationship between species richness and complementarity effects by N addition in the bivariate model (Fig. 2a). One possibility is that $eCO_2$ may mitigate the negative effects of N addition on legumes as $eCO_2$ can stimulate more net N mineralization and plant N pools in plots with high legume dominance than those with low legume dominance[28]. $eCO_2$ and N addition also interacted to strengthen the negative effect of species richness on selection effects (Fig. 2b), perhaps related to the increased dominance of C4 grasses with time in more diverse communities (Supplementary Fig. S5). $eCO_2$ may have particularly positive effects on the dominance of C4 grasses in enriched N conditions because $eCO_2$ can increase net N mineralization and soil N supply in C4-dominated communities[52].

Neither $eCO_2$, N addition or their interaction affected the impacts of species richness on species asynchrony or species stability (Fig. 2c, d; Fig. 4b–d), which is consistent with results from Hautier et al. [42,43]. Neither $eCO_2$ nor N addition affected the relationships between complementarity or selection effects and species asynchrony or species stability, which may result from the opposite effects of global change drivers on the relationships between complementarity or selection effects and species asynchrony at different species richness levels (Supplementary Fig. S3). In 4-species mixtures, N addition, $eCO_2$, and their interaction shifted the positive relationship between complementarity effects and species asynchrony to negative (Supplementary Fig. S3a). This suggests that in low-diversity communities, N addition and $eCO_2$ likely tended to decrease competition among functional groups (Supplementary Fig. S7), leading to lower species asynchrony (Supplementary Fig. S3a). In a 16-species mixture, N addition, $eCO_2$ and their interaction shifted the positive relationship between selection effects and species asynchrony to negative (Supplementary Fig. S3f). It suggests that in highly diverse communities, N addition and $eCO_2$ likely tended to increase the competition among dominant C4 grasses and other functional groups (Supplementary Fig. S7), as well as increase species asynchrony (Supplementary Fig. S3f).

While our study primarily focuses on aboveground productivity, plant diversity, and global change drivers can also affect community productivity via belowground processes, such as belowground resource partitioning[53]. Impacts of diversity and global change drivers on belowground productivity may be partly captured by aboveground responses, in case the above and belowground responses are coupled[54]. In addition, competition for light is an important driver of plant diversity loss under eutrophication, as shown by previous studies based on this experiment[55], further justifying the use of aboveground productivity to study community responses to global change drivers. It is possible that belowground productivity is more stable than aboveground productivity in response to environmental fluctuations due to the buffering effect of soils[56] and the greater complementarity in root depth for resource partitioning[57]. Future studies on how global change drivers affect the mechanisms between biodiversity-functioning and biodiversity-stability relationships of belowground productivity are needed to gain a more comprehensive understanding of ecosystem responses to global change. Moreover, our study is limited to a single grassland system, whereas extending research to inform policy and management requires more universally applicable conclusions. These relationships uncovered by our study could be different in natural ecosystems and at large spatial scales. For example, a meta-analysis of richness-manipulated experiments of terrestrial plants and aquatic algae showed that biodiversity can enhance both productivity and

stability, but the strength of the two effects is independent[8]. Another study based on natural ecosystems showed that there is no relationship between ecosystem diversity and their productivity and stability, while higher productivity is associated with greater stability[58].

In conclusion, our study investigated the effects of global change factors on biodiversity-functioning and biodiversity-stability relationships, and processes underlying the links between these relationships. We found that N addition and $eCO_2$ interacted to decrease the effects of biodiversity on complementarity and selection effects, but had no effect on biodiversity-stability relationships or the links between biodiversity-functioning and biodiversity-stability relationships. Our study offers important insights into the mechanisms that sustain ecosystem functioning and stability in the face of global environmental change.

## Methods

### Study sites and experimental design

Our analyses were based on the BioCON experiment (Biodiversity, $CO_2$ and N; "E141") at the Cedar Creek Ecosystem Science Reserve, Minnesota, United States (45°40′N, 93°18′W). The region has a continental climate with warmer summer (average temperature of 22 °C in July) and cold winter (average temperature of −11 °C in January), and average annual precipitation is 660 mm[59]. The soils are sandy (Typic Udipsamment, Nymore series; 94.4% sand, 2.5% clay). The experiment was established in 1997 on a secondary successional grassland after removing prior vegetation[59].

The BioCON manipulated biodiversity, $CO_2$, and N addition in a series of related experiments, all with a well-replicated split-plot design (Supplementary Fig. S8)[59,60]. The main random assemblage experiment contained 296 2 m × 2 m plots arranged in six circular 20-m diameter rings. Three rings were exposed to elevated $CO_2$ using free-air $CO_2$ enrichment (+180 ppm) and three to ambient $CO_2$. Within each ring, half of the plots received nitrogen (+4 g N m$^{-2}$ yr$^{-1}$ applied as ammonium nitrate). Plots at each of the four contrasting $CO_2$ and N levels were randomly assigned to four levels of plant species diversity (1, 4, 9, and 16 species) randomly chosen from a pool of 16 perennial grassland species. These species represent four functional groups, including C3 grasses (*Elymus repens* (formerly *Agropyron repens*), *Bromus inermis*, *Koeleria macrantha* (formerly *Koeleria cristata*), and *Poa pratensis*), C4 grasses (*Andropogon gerardii*, *Bouteloua gracilis*, *Schizachyrium scoparium,* and *Sorghastrum nutans*), legumes (*Amorpha canescens*, *Lespedeza capitata*, *Lupinus perennis*, and *Petalostemum villosum*), and non-legume forbs (*Achillea millefolium*, *Anemone cylindrica*, *Asclepias tuberosa,* and *Solidago rigida*). Additionally, to maintain the study site in a grassland state, the plots were burnt in spring for half of the years between 2000 and 2012 and every fall since 2013[61].

In August of each year from 1998 to 2021, aboveground biomass was collected by clipping a 10 cm × 100 cm strip in each plot just above the soil surface. The cover of each species was estimated in a fixed 50 cm × 100 cm quadrat in each plot. We used the data from 1998 to 2021 for the 296 plots in the main experiment. The dataset can be found at https://cedarcreek.umn.edu/research/data. A total of 48 out of 296 plots were used for additional experiments, with 24 plots for precipitation treatment from 2007 and 24 plots for precipitation ambient. Within each of the two precipitation levels, half of the plots received warming treatment from 2012[62]. Therefore, we excluded 24 plots with a precipitation reduction treatment from 2007 and another 12 plots with a warming treatment from 2012. Additionally, a few 4-species plots do not have cover or biomass data in 2020 due to limited sampling, and therefore were removed from the analysis for just that year.

We calculated biodiversity effects on productivity and its temporal stability based on aboveground biomass at both species and community levels. However, biomass collected in 9-species plots in

2005 and 2006, as well as in 4-species plots from 2006 to 2021, was unsorted into species-level. To fill the missing data (-11.98% of the total biomass dataset), we estimated the species-level biomass for these plots and years based on the cover data[39,46,61]. The details for estimation are as follows: (1) For the plots with both species-level biomass and cover, the mean biomass and mean cover of each species across years within each plot were calculated. Then linear models without intercepts were fitted to obtain the equations for the relationship between biomass and cover of each species, with the mean biomass of each species as the response variable and the mean cover of each species as the predictive variable. (2) The observed cover of each species was substituted into the equation to calculate the mean predicted biomass for each species in each plot. (3) The predicted biomass of each species was divided by the total predicted biomass of the community to obtain the predicted proportion of biomass of each species. (4) The predicted proportion of biomass of each species was multiplied by the observed total biomass of the plot to obtain the weighted predicted biomass of each species in each plot. We fitted the linear models to estimate the predictability of this method by comparing the weighted predicted biomass and the measured biomass for each species. Among the 16 equations we used, more than half of the species (10 in 16 species) were reliably predicted from their cover for productivity, with $R^2$ of the linear models larger than 0.5 (Supplementary Fig. S9). We conducted multiple sensitivity analyses (e.g., filling the biomass gaps or not, comparing different types of modeling approaches, and adding random noise to the predicted biomass; Supplementary Note 1) to determine that our results were not dependent on analytical choices.

## Community stability

We calculated community stability ($S_{com}$) as the sum of the temporal mean biomass of all species in a community ($Y_T$) divided by the sum of their temporal standard deviation ($\sigma_T$) over the 24-year period (1998–2021)[63] as:

$$S_{com} = \frac{Y_T}{\sigma_T} \tag{1}$$

$$Y_T = \sum_{i=1}^{N} Y_i \tag{2}$$

where $Y_i$ is the temporal biomass mean of species $i$ in the community. $N$ is the number of species planted in the mixture. Then we partitioned community stability into species stability and species asynchrony[21,64]. Species stability ($S_{sp}$) was calculated as the reciprocal of the weighted average of species-level temporal variability[24]:

$$S_{sp} = \frac{Y_T}{\sum_{i=1}^{N} \sigma_i} \tag{3}$$

Where $\sigma_i$ is the temporal biomass standard deviation of species $i$ in the community. And species asynchrony ($\varphi$) was calculated as[20]:

$$\varphi = \frac{\sum_{i=1}^{N} \sigma_i}{\sigma_T} \tag{4}$$

By definition,

$$S_{com} = S_{sp} \times \varphi \tag{5}$$

## Biodiversity effects on ecosystem functioning

We calculated net biodiversity effect (NBE, $\Delta Y$) and partitioned it into complementary effect (CE) and selection effect (SE) following Loreau and Hector[13]:

$$\Delta Y = \sum_{i=1}^{N} Y_i - \sum_{i=1}^{N} (RY_{e,i} \times M_i) = \sum_{i=1}^{N} \Delta RY_i \times M_i = CE + SE \tag{6}$$

$$\Delta RY_i = \frac{Y_i}{M_i} - RY_{e,i} \tag{7}$$

$$CE = N \times \overline{\Delta RY} \times \overline{M} \tag{8}$$

$$SE = N \times \text{cov}(\Delta RY, M) \tag{9}$$

In order to keep consistent with the calculations of community stability, here, we used the average biomass of species or communities over the 24 years[24]. Therefore, $Y_i$ is the average biomass of species $i$ in a mixture across years. $M_i$ is the average biomass of species $i$ in the monoculture across years. $RY_{e,i}$ is the expected relative yield of species $i$ in the mixture, which is simply the inverse of the number of species (1/$N$) in the plot because it was evenly seeded. We excluded species with a monoculture biomass less than $2.5\,\text{g}\,\text{m}^{-2}$ in a given plot and a given year because relative yield can approach infinity with small monoculture biomass values[65].

## Data analyses

To assess how nitrogen (N) addition and elevated $CO_2$ (e$CO_2$) affect the relationships between species richness and community productivity, and between species richness and community stability, we fitted linear mixed-effects models with mean of community productivity, community stability, complementarity effects, selection effects, species asynchrony or species stability as response variables, species richness, N addition, e$CO_2$ and their interactions as fixed effects, and rings as the random intercept (there is no strong correlation between the explanatory variables; Supplementary Fig. S10). Then we assessed how N addition and e$CO_2$ affected the relationships between the mean of community productivity and complementarity effects or selection effects, as well as the relationships between community stability and species asynchrony or species stability. We fitted linear mixed-effects models with the mean of community productivity or stability as response variable, and complementarity effects or selection effects, or species asynchrony or species stability and their interaction with N addition and e$CO_2$ as fixed effects, and rings as the random intercept. We also assessed how N addition and e$CO_2$ affected the relationships between the two partitions of biodiversity effects on productivity and those on community stability. We fitted linear mixed-effects models with species asynchrony or species stability as response variables, and complementarity effects or selection effects and their interactions with N addition and e$CO_2$ as fixed effects, and rings as the random intercept. We obtained $P$ values using the 'Anova' function in the car package[66] with type III sum of squares.

We used multi-group piecewise structural equation models (SEM) to test the relationship between biodiversity effects on community productivity and stability under different treatments (ambient, N addition, e$CO_2$, or the interaction between N addition and e$CO_2$). Based on an a priori model (Supplementary Fig. S11), we used linear mixed-effects models in SEM with rings as the random intercept. Specifically, we examined the effect of species richness on complementarity and selection effects that contribute to productivity. Similarly, we examined the effect of species richness on species asynchrony and species stability that contribute to community stability. We also assessed the effect of complementarity and selection effects on species asynchrony and species stability. Complementarity effect and selection effect, as well as species asynchrony and species stability, were considered as correlated error terms[67]. We simplified

the SEM by removing non-significant paths. Then, we performed a multi-group SEM analysis based on the global model to explore the dependence of these paths on different treatments. Paths were unconstrained if they differed significantly among different treatments, or they were kept constrained. The goodness-of-fit test of the models was assessed by Fisher's $C$ and $P$ value. All analyses were conducted in R version 4.4.0, using 'psem' and 'multi-group' functions in piecewiseSEM package for multi-group SEM[67], and the nlme package for linear mixed-effects models[68]. To improve the normality, stability, asynchrony, and species richness were log-transformed. When necessary, we used an exponential variance structure (varExp in the R package nlme) to improve homogeneity.

### Reporting summary

Further information on research design is available in the Nature Portfolio Reporting Summary linked to this article.

## Data availability

The plant aboveground biomass data and species percent cover data used in this paper are available at the figshare repository: https://doi.org/10.6084/m9.figshare.26841637.v3. The original data used in the data synthesis are available from data repositories: https://cedarcreek.umn.edu/research/data. The plant aboveground biomass data used in this study is available at https://portal.edirepository.org/nis/mapbrowse?packageid=knb-lter-cdr.302.newest. The plant species percent cover data is available at https://portal.edirepository.org/nis/mapbrowse?packageid=knb-lter-cdr.301.newest.

## Code availability

The main R code used in this study is available at the figshare repository: https://doi.org/10.6084/m9.figshare.26841637.v3.

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

## Acknowledgements

We thank all the people who contribute to the data collection and processing of BioCON experiments. This study was supported by National Natural Science Foundation of China (No. U22A20449 and 31830009 to S.Z.), by María de Maeztu Excellence Unit 2023-2027 (Ref. CEX2021-001201-M), funded by MCIN/AEI/10.13039/501100011033 and by grants from the US National Science Foundation ASCEND Biology Integration Institute (NSF-DBI-2021898), and Long-Term Ecological Research Program (LTER) including DEB-0620652, DEB-1234162 and DEB-1831944.

## Author contributions

M. Huang, Y.H., S.W., and P.B.R. conceived and designed the idea. P.B.R. contributed the data. M. Huang and Y.H. performed the analysis with contributions from P.B.R. and N.M. M. Huang and Y.H. wrote the manuscript with input from P.B.R., S.W., Y.F., P.H., K.E.B., M. He, S.L., S.Z., N.M., and F.I. All coauthors discussed the results and contributed to the final version.

## Competing interests

The authors declare no competing interests.
