## [Transparent Peer Review file · Nature Communications]

Nitrogen and CO₂ enrichment interact to decrease biodiversity impact on complementarity and selection effects

Corresponding Author: Dr Mengjiao Huang

Version 0:

Reviewer comments:

Reviewer #1

(Remarks to the Author)

Background dependency is a highly significant feature in the relationship between biodiversity, ecosystem functioning, and its stability (i.e., BEF&S). The underlying mechanisms are thought to involve how different environmental contexts modulate intermediate interactions, such as mutualism and competition, thereby influencing complementarity and selection effects—two primary mechanisms through which biodiversity enhances ecosystem function. This manuscript captures and delves deeply into this topic, presenting a timely, nuanced, and highly engaging study. I reviewed this manuscript with great interest, and I am thoroughly impressed with the quality of the data, statistical methods, results, and conclusions. I am pleased to recommend acceptance following the completion of major revisions. However, further improvement in academic writing is needed (see my line-by-line suggestions for specifics).

The authors utilized data from a 24-year experiment manipulating species richness, nitrogen addition, and elevated CO₂ levels. Their findings reveal that nitrogen addition and elevated CO₂—both core factors representing global change—tend to weaken the relationship between biodiversity and ecosystem functioning. Interestingly, however, these factors appear to have minimal impact on species stability and asynchrony. This insight contributes significantly to our understanding of how global change drivers may differentially affect biodiversity-function relationships and stability mechanisms, emphasizing the nuanced responses of ecosystems under changing environmental conditions. Below, I provide more specific suggestions to further enhance the quality of this work.

L35: I recommend rephrasing the sentence for clarity and simplicity to: “The Earth is undergoing an unprecedented decline in biodiversity due to global environmental changes, which significantly affect ecosystem functioning and stability. However, it remains unclear how these global changes interact to influence biodiversity-ecosystem functioning, biodiversity-ecosystem stability relationships, and the connections between these processes.”

L39: Please rephrase this sentence to emphasize the research objective more directly, with the data as a supporting tool: “Here, we investigate the impacts of nitrogen (N) addition, enriched CO₂ (eCO₂), and their interaction on the links between complementarity effects, selection effects, species asynchrony, and species stability, using data from a long-term, 24-year experiment on biodiversity and global change.”?

L52: Please check the word count of the abstract, as it may exceed the requirements set by the journal's guidelines.

L63: This sentence is a bit unclear; please consider rephrasing and clarifying. “...it is crucial to understand the processes driving biodiversity loss in response to global changes, as well as the resulting consequences for ecosystem functioning and stability”?

L78-85: It seems you're aiming to emphasize the importance of clarifying the relationship between function and stability (synergy vs. trade-offs), which is indeed a valuable focus. However, this is already the third paragraph of your introduction! Where is the knowledge gap? The introduction should remain concise, especially for high-impact journals, where brevity is essential. You might even consider moving this section to the discussion. The introduction should quickly highlight the current research gap and the specific focus of your study. I suggest restructuring the entire introduction with this in mind.

L92: Excellent! Including a summary table is very helpful!

L103-105: This statement is excellent! 'If global changes lead to a disconnection between the processes that enhance productivity and stability, it could result in ecosystems that are highly productive but unstable, or stable but with reduced productivity.' However, it would be beneficial to discuss additional findings in this area. Currently, most manipulative experiments focus on a single ecosystem type, such as the commonly studied grasslands. Yet, natural ecosystems are complex, and, on a global scale—across forests, grasslands, and sparse grasslands—higher productivity is generally associated with greater stability (Fernández-Martínez et al. 2020). Even if a decoupling of productivity and stability is observed within a single grassland type, this conclusion becomes more debatable when considering larger scales that encompass a diversity of ecosystem types.

Fernández-Martínez, M., Sardans, J., Musavi, T., Migliavacca, M., Iturrate-García, M., Scholes, R.J. et al. (2020). The role of climate, foliar stoichiometry and plant diversity on ecosystem carbon balance. *Global Change Biology*, 26, 7067-7078.

L113: "Nitrogen detriment hypothesis"? Nitrogen addition may also lead to soil acidification or even toxicity, altering the composition of microbial communities

L132-134: Excellent job identifying the knowledge gap. However, this is already the eighth paragraph of the introduction!

L155: "not necessarily"? It seems imprecise for objectively reporting results and may need rephrasing for clarity.

L163: It's generally believed that greater niche differentiation leads to stronger compensatory dynamics and species asynchrony. However, in this case, the latter two show no association. Why might this be? Additionally, what is the observed relationship between productivity and stability under the environmental conditions in this study?

L182: 'marginally positive'? Please review the entire Result section for statistical significance reporting. You've used frequentist methods, but haven't consistently noted statistical significance, which could lead to confusion. If you were using Bayesian analysis, moving beyond p-values would be commendable; however, in this case, please include p-values where relevant.

L190-193: Maybe "In this study, using data from the long-term BioCON experiment manipulating biodiversity and global change, we quantified the impact of two global change factors—elevated CO₂ (eCO₂) and nitrogen (N) addition—and their interaction on biodiversity-function relationships, biodiversity-ecosystem stability relationships, and the links between these relationships."??

L216-219: Are you suggesting here that species richness drives community stability through asynchrony rather than species stability? If so, please consider rephrasing this sentence for clarity.

L226-227: So, species richness modulates the relationship between complementarity effects and species asynchrony? Very interesting explanation.

Line 243: Could a reduction in competition decrease complementarity effects?

Line 262: The lack of significant effects may be a promising sign. However, I suggest adding more discussion on potential uncertainties here. Extending BEF&S research to inform policy and management requires more universally applicable conclusions. While I understand that long-term, single-site observations are invaluable, it's also important to acknowledge the challenges of generalizing findings from a single location to other ecosystems.

Methods

L403, L413-415: Have you checked for multicollinearity among the explanatory variables in your models? Including interaction terms often increases the risk of high multicollinearity between variables, which can impact the robustness of parameter estimates.

L428: Excellent use of robust statistical methods!

L640: It seems that the multi-group structural equation model only reports an overall model fit (Fisher's C and P-value), which I find a bit confusing. In my experience, including my own use of PSEM analysis with grouped data (e.g., fitting by ecosystem type), it is standard practice to report the fit for each group separately.

L646-647: 'Non-significant paths are not shown (P ≥ 0.1).' Shouldn't non-significant paths be indicated as P > 0.05 instead?

(Remarks on code availability)

Thanks for sharing these codes! Looks great.

Reviewer #2

(Remarks to the Author)

The study of Huang et al. aims at exploring the impacts of N addition and CO₂ enrichment on the biodiversity-productivity relationship, biodiversity-stability relationship, as well as the links between complementarity effects, selection effects, species asynchrony and species stability. The authors make use of an extensive dataset from a long-term biodiversity and global change experiment conducted in Minnesota, United States. The results show that species diversity increase both community productivity and stability under N addition and CO₂ enrichment. Nevertheless, N addition and CO₂ enrichment diminished the positive impacts of biodiversity on complementarity and selection effects. In contrast, N addition and CO₂ enrichment did not alter the relationship between biodiversity and species asynchrony or stability. The topic is interesting and the manuscript is well-written and systematically presented. However, it still has several flaws, which I outline below.

Major comments:

1. Clarity on Experimental Design and Plot Details: While the authors provide some context and references regarding the experimental setup, critical details about the experimental plots remain unclear. Specifically, it is important to know how many plots were included in each treatment and across different levels of species richness, how these plots were distributed among different experimental rings, and whether individual productivity was influenced by edge effects (i.e., whether the growth of individuals at the plot's edge was impacted by the diversity of adjacent plots as well as the plot they were located in). I recommend that the authors include additional clarifications or visual aids, such as diagrams, either in the main text or supplementary information.

2. Incomplete Addressing of the Study's Aims: A central concern is that, although the authors state that a primary objective of the paper is to explore how biodiversity influences productivity and stability via complementarity effects, selection effects, species asynchrony, and species stability, this question is only partially addressed. The current structural equation model (SEM; Figure 4) does not sufficiently reflect how these processes mediate the biodiversity-productivity and biodiversity-stability relationships. I strongly recommend that the authors revise their SEM to incorporate productivity and stability explicitly. This revision would allow for a more comprehensive analysis of how different treatments (ambient, N addition, CO₂ enrichment, and the N-CO₂ interaction) affect these relationships, and the authors should discuss these impacts based on the updated model results.

Additional specific comments:

L81-85 and L263-266: That being said, the findings do not capture the trade-off or balance between community productivity and stability.

L124-134: the explanation of the impact of N addition on biodiversity and productivity is clear, but the explanation about CO₂ enrichment, and the N-CO₂ interaction is not sufficient.

L140-148, I suggest that the author double check the conformity between these questions raised and the presented SEM model (Figure 4 and S9).

(Remarks on code availability)

The author provides clear and executable code

Reviewer #3

(Remarks to the Author)

This manuscript presents a valuable study; linking global change effects of biodiversity interaction/function mechanisms. Using a long established global change experiment, it interprets two drivers (N and CO₂) and four responses (complementarity, selection, species asynchrony and selection effects). Inherently it is a complex study which the authors explain relatively clearly. However I am concerned with two aspects of the production data.

Firstly and most importantly, the predictive model for aboveground biomass does not perform particularly well for many of the species. The 0.5 threshold given in the paper is a minimum and many of the models do not perform much better than this. Given the large amount of reliance on this implied biomass data the errors in these models surely have a major effect on the relationships found in between the global change factors and biodiversity /function relationships. The poor fitting models are, as far as I can tell, still included in the biomass predictions. Obviously these data cannot be retroactively produced but propagating errors through may give insight on which of the fundamental relationships can really be trusted. All the interpretation on the paper relies on this, which makes me wonder if its fundamentally shaky or not. The authors must address this as the potential for over-interpreting correlations on this modelled data is high which means the impactful results could be completely wrong!

This first point also raises a sub-issue: the large numbers of linear models in the paper on non-independent data surely run the risk of Type I errors via multiple comparisons errors arising. This should be accounted for.

Secondly both the global change factors affect both above and belowground parts of plants. Again such data are impossible to gather in retrospect, but this is not mentioned at all in the paper and I wonder how many of the effects on productivity would be mitigated by shifts in above-belowground ratios, particularly N addition as the herbaceous communities studied would be potentially N limited. This is just part of how the discussion could be improved, which in general is written in an

overly systematic way which does not discuss the results synthetically.

I also find the SEM part of the paper somewhat confusing. The method would be easier to follow with references to the piecewise SEM method in a publication. I was however happy to see that the starting model is easy to understand and straightforwardly structured.

Specific comments

L43 – this line in the abstract is quite confusing as productivity and stability of productivity have not been mentioned until now

L59 – but these are not the biggest drivers of decline?

L63 – this reads like productivity is the only function we care about from ecosystems, clearly not the case

L78 – this line is overly complex for the implied meaning.

(I stopped writing comments on the discussion as this may change based on model updates)

(Remarks on code availability)

The paper results seem reproducible from the code.

Version 1:

Reviewer comments:

Reviewer #1

(Remarks to the Author)

I appreciate the authors' efforts in addressing my comments. The quality of the manuscript has significantly improved compared to the initial version. Therefore, I recommend its acceptance.

(Remarks on code availability)

The code looks well-structured; however, I could not find the necessary data (or perhaps I overlooked something?), so I was unable to run it successfully. I would appreciate it if the authors could provide the required data for analysis.

Reviewer #2

(Remarks to the Author)

I have carefully read the author's responses to my comments and also read the new version of the paper. I am now satisfied with this new version. I am happy to accept this paper now without any further corrections.

(Remarks on code availability)

The author provides clear and executable code

Reviewer #3

(Remarks to the Author)

I still think this is a noteworthy paper on an interesting topic. I am generally happy with the authors' revision and their rebuttal to the issues with the models that I raised in my previous review, although I think a little of this information – in particular the small fraction of the total dataset from predictive models – should be clear in the text as my initial criticism is less valid with this extra knowledge only available in the comments review.

However, I still have an issue with the discussion of belowground effects. Specifically, this is because this is a paper about disentangling effects which are all fundamentally about competition. If most of this competition occurs belowground – this is quite well established in the literature - and above and belowground biomass partitioning is plastic – also well established - , can we really be confident about the relative importance and response of such effects from only aboveground data? I appreciate that such measurements are more difficult and cannot be conducted post hoc, but I think the authors' text addition misses the point as it doesn't matter if generally N addition shifts allocation above ground (notably eCO₂ is not mentioned here and may be the other direction - this should be addressed); even if allometric shifts under global change favour above ground this may still mean competition happens belowground!

Minor comments

L155 – this line ends abruptly

The new supplementals are useful for understanding the assumptions and reliability of the study but, the new figures and text are not always clear. I am not sure what “Most species including species richness = 1 have a good model fit ($R^2 > 0.7$)” means, when models are either all species level or species richness > 1

There is also a typo in “Not: we ran it separately because it took a long time.”

(Remarks on code availability)

Reviewer #1 (Remarks on code availability):

Thanks for sharing these codes! Looks great.

> Thank you for your positive feedback to the codes. We updated the codes for this new version. Please see <https://figshare.com/s/86a6a4d9c5cda09a6c0c>.

Reviewer #2 (Remarks to the Author):

The study of Huang et al. aims at exploring the impacts of N addition and CO₂ enrichment on the biodiversity-productivity relationship, biodiversity-stability relationship, as well as the links between complementarity effects, selection effects, species asynchrony and species stability. The authors make use of an extensive dataset from a long-term biodiversity and global change experiment conducted in Minnesota, United States. The results show that species diversity increase both community productivity and stability under N addition and CO₂ enrichment. Nevertheless, N addition and CO₂ enrichment diminished the positive impacts of biodiversity on complementarity and selection effects. In contrast, N addition and CO₂ enrichment did not alter the relationship between biodiversity and species asynchrony or stability. The topic is interesting and the manuscript is well-written and systematically presented. However, it still has several flaws, which I outline below.

Major comments:

1. Clarity on Experimental Design and Plot Details: While the authors provide some context and references regarding the experimental setup, critical details about the experimental plots remain unclear. Specifically, it is important to know how many plots were included in each treatment and across different levels of species richness, how these plots were distributed among different experimental rings, and whether individual productivity was influenced by edge effects (i.e., whether the growth of individuals at the plot's edge was impacted by the diversity of adjacent plots as well as the plot they were located in). I recommend that the authors include additional clarifications or visual aids, such as diagrams, either in the main text or supplementary information.

> Thank you for this suggestion. We added a figure in Supplementary Fig. S8 to show the aerial view of one ring, the number of plots in each CO₂ and nitrogen treatment, and each species richness level. The plots in each ring were randomly assigned to nitrogen treatment and species richness level. To avoid potential edge effects, sampling strips were located at a distance away from plot edges. We added more details to the revised method in the text.

2. Incomplete Addressing of the Study's Aims: A central concern is that, although the authors state that a primary objective of the paper is to explore how biodiversity influences productivity and stability via complementarity effects, selection effects, species asynchrony, and species stability, this question is only partially addressed. The current structural equation model (SEM; Figure 4) does not sufficiently reflect how these processes mediate the biodiversity-productivity and biodiversity-stability relationships. I strongly recommend that the authors

revise their SEM to incorporate productivity and stability explicitly. This revision would allow for a more comprehensive analysis of how different treatments (ambient, N addition, CO₂ enrichment, and the N-CO₂ interaction) affect these relationships, and the authors should discuss these impacts based on the updated model results.

> Thank you for this suggestion. We incorporated productivity and community stability in the updated SEM in the revised Fig. 4. Specifically, we considered the relationship between complementarity and selection effects with productivity. Community stability is linked to species asynchrony and species stability, while species stability is also linked to productivity (Wagg et al. 2022). We also modified the prior SEM (Fig. S11) and the related results in lines 149-166. Our main conclusion does not change, i.e., 'N addition and eCO₂ together diminished the effects of biodiversity on complementarity and selection effects. In contrast, N addition and eCO₂ did not alter the relationship between biodiversity and species asynchrony or stability. The links between complementarity effects and selection effects with species asynchrony and species stability persisted under N addition and eCO₂.'

Additional specific comments:

L81-85 and L263-266: That being said, the findings do not capture the trade-off or balance between community productivity and stability.

> We deleted the previous L81-85 according to Reviewer #1' comment about restructuring and simplifying the entire introduction. Regarding the previous L263-266, we agree with the reviewer and deleted 'the balance between' in this sentence.

L124-134: the explanation of the impact of N addition on biodiversity and productivity is clear, but the explanation about CO₂ enrichment, and the N-CO₂ interaction is not sufficient.

> To our knowledge, there is few evidence about how CO₂ enrichment, and the N-CO₂ interaction affect biodiversity-functioning and biodiversity-stability relationships. As we mentioned in the text, 'A combination of eCO₂ and N addition increased productivity more than eCO₂ alone. However, whether eCO₂ and N addition interact to affect the processes underlying biodiversity-stability relationships, as well as biodiversity-functioning and biodiversity-stability relationships remain unknown.' This is also one of our research questions in this study.

L140-148, I suggest that the author double check the conformity between these questions raised and the presented SEM model (Figure 4 and S9).

> As also mentioned above, we incorporated productivity and community stability into the updated SEM in Fig. 4 to better align with our research questions. Specifically, we included productivity and community stability in the SEM to show the pathways through which species richness affects productivity (via complementarity effect or selection effect) and stability (via species asynchrony or species stability). For the treatment impact, we focus on how N addition and eCO₂ affect the processes underlying BEF (i.e., the relationship between diversity

and complementarity and selection effects) and BEFS relationships (i.e., the relationship between diversity and species asynchrony and species stability), as we addressed in the research questions and the Introduction.

Reviewer #2 (Remarks on code availability):

The author provides clear and executable code

> Thank you for your positive feedback to the codes. We updated the codes for this new version. Please see <https://figshare.com/s/86a6a4d9c5cda09a6c0c>.

Reviewer #3 (Remarks to the Author):

This manuscript presents a valuable study; linking global change effects of biodiversity interaction/function mechanisms. Using a long established global change experiment, it interprets two drivers (N and CO₂) and four responses (complementarity, selection, species asynchrony and selection effects). Inherently it is a complex study which the authors explain relatively clearly.

However I am concerned with two aspects of the production data.

Firstly and most importantly, the predictive model for aboveground biomass does not perform particularly well for many of the species. The 0.5 threshold given in the paper is a minimum and many of the models do not perform much better than this. Given the large amount of reliance on this implied biomass data the errors in these models surely have a major effect on the relationships found in between the global change factors and biodiversity /function relationships. The poor fitting models are, as far as I can tell, still included in the biomass predictions. Obviously these data cannot be retroactively produced but propagating errors through may give insight on which of the fundamental relationships can really be trusted. All the interpretation on the paper relies on this, which makes me wonder if its fundamentally shaky or not. The authors must address this as the potential for over-interpreting correlations on this modelled data is high which means the impactful results could be completely wrong!

> We agree that the biomass prediction models do not perform well for some species. We first compared the predictive power of different types of modelling approaches (e.g. exponential model, power-law model) and found that our original linear regressions performed the best (i.e. having the largest R²) and there is no significant difference among different modelling approaches. We found that species had smaller predictive power if they were frequently present in the biomass dataset but frequently absent in the cover dataset, or vice versa.

To investigate the impacts of the inclusion of predicted biomass (11.98% of the whole dataset) on our results, we then re-ran all our analyses using different approaches and compared the results. Specifically,

1. we compared the results when predicted biomass data was included vs excluded. We found that, despite some variability, the overall patterns of the results excluding the predicted biomass align with those presented in Figs. 1–4 of the main text. From the SEM, we can still get the conclusion that global change decreased the relationship between species richness and complementary effect, and have no significant effect on the links between BEF and BEFS relationships.

2. we also compared the results when random noise drawn from the fitted regression models was added to the predicted biomass or not. The results after adding random noise are very similar to the original results in Figs. 1–4 of the main text. These additional analyses demonstrate that the inclusion of predicted biomass had very limited effects on our results. This may be unsurprising given the predicted biomass accounted only for a small proportion (11.98%) of whole dataset. Lastly, the approach has been supported by previous studies using the BioCON dataset (e.g., Reich et al. 2012; Isbell et al. 2013; Mohanbabu et al. 2024).

We added this information to the method section in l. 444-447 and also in Supplementary Methods.

This first point also raises a sub-issue: the large numbers of linear models in the paper on non-independent data surely run the risk of Type I errors via multiple comparisons errors arising. This should be accounted for.

> We fitted the linear model for each species for the relationship between observed biomass and observed cover. We used the R^2 to check the prediction ability, and the regression coefficient for biomass prediction. To our knowledge, Type I errors typically arise when related to the tests of significance. We didn't use the p values of these models. Therefore, we don't think we are risking Type I errors regarding these linear models for biomass prediction.

Secondly both the global change factors affect both above and belowground parts of plants. Again such data are impossible to gather in retrospect, but this is not mentioned at all in the paper and I wonder how many of the effects on productivity would be mitigated by shifts in above-belowground ratios, particularly N addition as the herbaceous communities studied would be potentially N limited. This is just part of how the discussion could be improved, which in general is written in an overly systematic way which does not discuss the results synthetically.

> We added the discussion about how global changes, e.g., N addition, affect belowground productivity and its stability in l. 353-362. Generally, 'previous studies showed that plants had larger responses in above-than belowground under N addition because they allocate relatively more biomass aboveground (Gao et al., 2011). Evidence also showed that belowground productivity is more stable than aboveground productivity in response to environment fluctuations due to the buffering effect of soils (Yang et al., 2022) and the greater complementarity in root depth for resource partitioning (Xu et al. 2024). However, how global changes affect the mechanisms between biodiversity-functioning and biodiversity-stability

relationships of belowground productivity remain unclear’.

I also find the SEM part of the paper somewhat confusing. The method would be easier to follow with references to the piecewise SEM method in a publication. I was however happy to see that the starting model is easy to understand and straightforwardly structured.

> Thank you for your suggestion. We added more details to the SEM processes to be clearer.

Specific comments

L43 – this line in the abstract is quite confusing as productivity and stability of productivity have not been mentioned until now

> We revised the sentence in l. 50 and mentioned ecosystem productivity and its stability to be clearer.

L59 – but these are not the biggest drivers of decline?

> We agree that nitrogen addition and CO₂ enrichment are not the biggest drivers of biodiversity loss. According to Jaureguiberry et al. (2022), the dominant direct driver of recent biodiversity loss worldwide is land/sea use change. We removed the examples in this sentence to avoid confusion.

L63 – this reads like productivity is the only function we care about from ecosystems, clearly not the case

> Similarly, we removed the example in this sentence to avoid confusion.

L78 – this line is overly complex for the implied meaning.

(I stopped writing comments on the discussion as this may change based on model updates)

> We simplified this sentence as ‘These bipartite frameworks help clarify the links between biodiversity-functioning and biodiversity-stability relationships, enhancing our knowledge of biodiversity theory and its application to ecosystem management’.

Reviewer #3 (Remarks on code availability):

The paper results seem reproducible from the code.

> Thank you for your positive feedback to the codes. We updated the codes for this new version. Please see <https://figshare.com/s/86a6a4d9c5cda09a6c0c>.

Reference

Barry, K. E., Mommer, L., van Ruijven, J., Wirth, C., Wright, A. J., Bai, Y., ... & Weigelt, A. (2019). The future of complementarity: disentangling causes from consequences. Trends

in ecology & evolution, 34(2), 167-180.

- Craven, D., Isbell, F., Manning, P., Connolly, J., Bruelheide, H., Ebeling, A., ... & Eisenhauer, N. (2016). Plant diversity effects on grassland productivity are robust to both nutrient enrichment and drought. *Philosophical Transactions of the Royal Society B: Biological Sciences*, 371(1694), 20150277.
- Gao, Y. Z., Chen, Q., Lin, S., Giese, M., & Brueck, H. (2011). Resource manipulation effects on net primary production, biomass allocation and rain-use efficiency of two semiarid grassland sites in Inner Mongolia, China. *Oecologia*, 165, 855-864.
- Isbell, F., Reich, P. B., Tilman, D., Hobbie, S. E., Polasky, S., & Binder, S. (2013). Nutrient enrichment, biodiversity loss, and consequent declines in ecosystem productivity. *Proceedings of the National Academy of Sciences*, 110(29), 11911-11916.
- Jaureguiberry, P., Titeux, N., Wiemers, M., Bowler, D. E., Coscieme, L., Golden, A. S., ... & Purvis, A. (2022). The direct drivers of recent global anthropogenic biodiversity loss. *Science advances*, 8(45), eabm9982.
- Loreau, M., & Hector, A. (2001). Partitioning selection and complementarity in biodiversity experiments. *Nature*, 412(6842), 72-76.
- Mohanbabu, N., Isbell, F., Hobbie, S. E., & Reich, P. B. (2024). Species interactions amplify functional group responses to elevated CO₂ and N enrichment in a 24-year grassland experiment. *Global Change Biology*, 30(8), e17476.
- Reich, P. B., Tilman, D., Isbell, F., Mueller, K., Hobbie, S. E., Flynn, D. F., & Eisenhauer, N. (2012). Impacts of biodiversity loss escalate through time as redundancy fades. *Science*, 336(6081), 589-592.
- Wagg, C., Roscher, C., Weigelt, A., Vogel, A., Ebeling, A., De Luca, E., ... & Schmid, B. (2022). Biodiversity–stability relationships strengthen over time in a long-term grassland experiment. *Nature communications*, 13(1), 7752.
- Xu, Z., Jiang, L., Ren, H., & Han, X. (2024). Opposing responses of temporal stability of aboveground and belowground net primary productivity to water and nitrogen enrichment in a temperate grassland. *Global Change Biology*, 30(1), e17071.
- Yan, Y., Connolly, J., Liang, M., Jiang, L., & Wang, S. (2021). Mechanistic links between biodiversity effects on ecosystem functioning and stability in a multi-site grassland experiment. *Journal of Ecology*, 109(9), 3370-3378.
- Yang, G. J., Hautier, Y., Zhang, Z. J., Lü, X. T., & Han, X. G. (2022). Decoupled responses of above-and below-ground stability of productivity to nitrogen addition at the local and larger spatial scale. *Global Change Biology*, 28(8), 2711-2720.

REVIEWERS' COMMENTS

Reviewer #1 (Remarks to the Author):

I appreciate the authors' efforts in addressing my comments. The quality of the manuscript has significantly improved compared to the initial version. Therefore, I recommend its acceptance.

> We are grateful that you are satisfied with the new version.

Reviewer #1 (Remarks on code availability):

The code looks well-structured; however, I could not find the necessary data (or perhaps I overlooked something?), so I was unable to run it successfully. I would appreciate it if the authors could provide the required data for analysis.

> Thank you for this suggestion. We have uploaded the data to the figshare repository: <https://doi.org/10.6084/m9.figshare.26841637.v2>. We added this information in 'Data availability' section.

Reviewer #2 (Remarks to the Author):

I have carefully read the author's responses to my comments and also read the new version of the paper.

I am now satisfied with this new version. I am happy to accept this paper now without any further corrections.

> We are grateful that you are satisfied with the new version.

Reviewer #2 (Remarks on code availability):

The author provides clear and executable code

> Thank you for the positive feedback on the code.

Reviewer #3 (Remarks to the Author):

I still think this is a noteworthy paper on an interesting topic. I am generally happy with the authors revision and their rebuttal to the issues with the models that I raised in my previous review, although I think a little of this information – in particular the small fraction of the

total dataset from predictive models – should be clear in the text as my initial criticism is less valid with this extra knowledge only available in the comments review.

> Good point, we have now added the information that the predicted biomass is a small fraction of the total dataset in the Supplementary Note 1 and in the Method section in the main text in L339-340.

However, I still have an issue with the discussion of belowground effects. Specifically, this is because this is a paper about disentangling effects which are all fundamentally about competition. If most of this competition occurs belowground – this is quite well established in the literature - and above and belowground biomass partitioning is plastic – also well established, can we really be confident about the relative importance and response of such effects from only aboveground data? I appreciate that such measurements are more difficult and cannot be conducted post hoc, but I think the authors text addition misses the point as it doesn't matter if generally N addition shifts allocation above ground (notably eCO₂ is not mentioned here and may be the other direction - this should be addressed); even if allometric shifts under global change favour above ground this may still mean competition happens belowground!

> We agree with the reviewer that competition happens both aboveground and belowground. In this study, we focused mainly on aboveground productivity because (1) aboveground competition, including competition for light, has been identified as the main driver of community compositional change in response to eutrophication in this experiment (Reich et al. 2024, Nature), and (2) belowground competition should also be reflected by aboveground productivity. We have discussed the potential limitation of our study and added suggestion for future studies.

Minor comments

L155 – this line ends abruptly

> In L112 (previous L155), we revised the sentence as ‘...as well as the links between their underlying processes by asking how N addition and eCO₂ impact...’ to be clearer.

The new supplementals are useful for understanding the assumptions and reliability of the study but, the new figures and text are no always clear. I am not sure what “Most species including species richness = 1 have a good model fit ($R^2 > 0.7$)” means, when models are either all species level or species richness > 1

> We have revised this sentence as ‘These relationships were evaluated both across all species richness levels (shown in black) and excluding monocultures (species richness > 1,

shown in blue). Most species—with the dataset across all species richness levels (shown in black)—showed good model fits ($R^2 > 0.7$). We also revised the Supplementary Note 1 thoroughly to improve its readability.

There is also a typo in “Not: we ran it separately because it took a long time.”

> Revised.

The study of Huang et al. aims at exploring the impacts of N addition and CO₂ enrichment on the biodiversity-productivity relationship, biodiversity-stability relationship, as well as the links between complementarity effects, selection effects, species asynchrony and species stability. The authors make use of an extensive dataset from a long-term biodiversity and global change experiment conducted in Minnesota, United States. The results show that species diversity increase both community productivity and stability under N addition and CO₂ enrichment. Nevertheless, N addition and CO₂ enrichment diminished the positive impacts of biodiversity on complementarity and selection effects. In contrast, N addition and CO₂ enrichment did not alter the relationship between biodiversity and species asynchrony or stability. The topic is interesting and the manuscript is well-written and systematically presented. However, it still has several flaws, which I outline below.

Major comments:

1. **Clarity on Experimental Design and Plot Details:** While the authors provide some context and references regarding the experimental setup, critical details about the experimental plots remain unclear. Specifically, it is important to know how many plots were included in each treatment and across different levels of species richness, how these plots were distributed among different experimental rings, and whether individual productivity was influenced by edge effects (i.e., whether the growth of individuals at the plot's edge was impacted by the diversity of adjacent plots as well as the plot they were located in). I recommend that the authors include additional clarifications or visual aids, such as diagrams, either in the main text or supplementary information.
2. **Incomplete Addressing of the Study's Aims:** A central concern is that, although the authors state that a primary objective of the paper is to explore how biodiversity influences productivity and stability via complementarity effects, selection effects, species asynchrony, and species stability, this question is only partially addressed. The current structural equation model (SEM; Figure 4) does not sufficiently reflect how these processes mediate the biodiversity-productivity and biodiversity-stability relationships. I strongly recommend that the authors revise their SEM to incorporate productivity and stability explicitly. This revision would allow for a more comprehensive analysis of how different treatments (ambient, N addition, CO₂ enrichment, and the N-CO₂ interaction) affect these relationships, and the authors should discuss these impacts based on the updated model results.

Additional specific comments:

L81-85 and L263-266: That being said, the findings do not capture the trade-off or balance between community productivity and stability.

L124-134: the explanation of the impact of N addition on biodiversity and productivity is clear, but the explanation about CO₂ enrichment, and the N-CO₂ interaction is not sufficient.

L140-148, I suggest that the author double check the conformity between these questions raised and the presented SEM model (Figure 4 and S9).